# Cell-Population Dynamics in Diffuse Gliomas during Gliomagenesis and Its Impact on Patient Survival

**DOI:** 10.3390/cancers15010145

**Published:** 2022-12-26

**Authors:** Pavel V. Nikitin, Guzel R. Musina, Stanislav I. Pekov, Andrey A. Kuzin, Igor A. Popov, Artem Y. Belyaev, Gregory L. Kobyakov, Dmitry Y. Usachev, Viktor N. Nikolaev, Valentin P. Mikhailov

**Affiliations:** 1University of Oslo, 0315 Oslo, Norway; 2Institute for Regenerative Medicine, Sechenov First Moscow State Medical University, 119991 Moscow, Russia; 3Center for Brain and Neurotechnology, 117513 Moscow, Russia; 4Skolkovo Institute of Science and Technology, 121205 Moscow, Russia; 5Siberian State Medical University, 634050 Tomsk, Russia; 6Moscow Institute of Physics and Technology, 141701 Dolgoprudny, Russia; 7Burdenko Neurosurgical Center, 125047 Moscow, Russia; 8N.N. Blokhin Cancer Research Center (RCRC), 115522 Moscow, Russia

**Keywords:** cell populations, intratumoral heterogeneity, astrocytoma, oligodendroglioma, glioblastoma, glioma stem cells

## Abstract

**Simple Summary:**

There is only a relatively small set of tools for the diagnosis and therapy of diffuse gliomas. To solve this problem, intratumoral heterogeneity has become the most important research topic. In this study, through the prism of cell-population heterogeneity in diffuse gliomas, using flow cytometry, multimeric FISH, immunofluorescence, PCR, and cell cultures, we comparatively examined the dynamics of cell populations’ quantitative and qualitative transformations in diffuse gliomas. For the first time, we identified mesenchymal-like cellular transformation in astrocytomas and oligodendrogliomas as possibly the key mechanism of malignancy. In the case of glioblastoma, it was shown for the first time that mesenchymal transformation is of a deeper nature, passing through the stem-cell link. The transformation severity influenced the prognosis of overall and relapse-free patient survival.

**Abstract:**

Diffuse gliomas continue to be an important problem in neuro-oncology. To solve it, studies have considered the issues of molecular pathogenesis from the intratumoral heterogeneity point. Here, we carried out a comparative dynamic analysis of the different cell populations’ content in diffuse gliomas of different molecular profiles and grades, considering the cell populations’ functional properties and the relationship with patient survival, using flow cytometry, immunofluorescence, multiparametric fluorescent in situ hybridization, polymerase chain reaction, and cultural methods. It was shown that an increase in the IDH-mutant astrocytomas and oligodendrogliomas malignancy is accompanied by an increase in stem cells’ proportion and mesenchymal cell populations’ appearance arising from oligodendrocyte-progenitor-like cells with cell plasticity and cells’ hypoxia response programs’ activation. In glioblastomas, malignancy increase is accompanied by an increase in both stem and definitive cells with mesenchymal differentiation, while proneuronal glioma stem cells are the most likely the source of mesenchymal glioma stem cells, which, in hypoxic conditions, further give rise to mesenchymal-like cells. Clinical confirmation was a mesenchymal-like cell and mesenchymal glioma stem cell number, and the hypoxic and plastic molecular programs’ activation degree had a significant effect on relapse-free and overall survival. In general, we built a multi-vector model of diffuse gliomas’ pathogenetic tracing up to the practical plane.

## 1. Introduction

To date, diffuse gliomas continue to be an important unsolved problem in neuro-oncology. Despite significant progress in understanding the molecular basis of the development of this diseases group, which has brought considerable success in a number of other oncology areas and has made it possible to create diagnostic and therapeutic approaches that are fundamentally new in their effectiveness, the improvements achieved in the case of diffuse gliomas seem to be very modest [1,2,3,4]. In particular, despite some improvement in diagnostic processes achieved due to the widespread molecular diagnostic criteria introduction, including IDH1 and IDH2 mutational status, as well as the 1p/19q co-deletion, TERT promoter mutations, EGFR mutations, chromosome 7 additions and of chromosome 10 loss, MGMT gene promoter methylation, and some others, there is a fairly large field for potential improvement of the diagnostic process [5,6]. Achievements in the development of targeted therapy for diffuse gliomas are currently considered to be very small [7,8]. In this regard, the need for a deeper fundamental study of different diffuse glioma forms becomes obvious, especially in regard to the comparative aspect of two fundamentally different molecular lines of these neoplasms’ development, namely diffuse gliomas with IDH1/IDH2 mutation (IDH-mut) and glioblastomas without mutations in these genes [9,10].

To solve this problem, some scientific groups have resorted to considering the issues of molecular pathogenesis from the point of view of intratumoral rather than intertumoral heterogeneity. In particular, in relation to diffuse gliomas, IDH-mut, it was shown by using single-cell RNA sequencing that astrocytomas and oligodendrogliomas are functionally similar to a large extent: populations of the so-called astrocyte-like cells (AC-like) and oligodendrocyte-progenitor-like cells (OPC-like) have almost parity content. Moreover, the malignization of these neoplasms is also accompanied, regardless of the histogenetic variant, by fairly similar changes, consisting of an increase in the number of stem cells [11,12,13,14]. Based on the results of single-cell RNA sequencing, glioblastoma is characterized by a significantly more variegated picture in which, in addition to the previously presented AC-like and OPC-like cell populations, one may distinguish a neural progenitor-like cell population (NPC-like) and a mesenchymal-like cell population (MES-like) that is subdivided into two subpopulations—MES1, hypoxia-independent, and MES2, hypoxia-dependent. Specific markers have been found for each tumor cell population that can be used for their identification and characterization. It should be noted that these markers are not absolute in the dynamic plane since they may not always adequately keep up with the rapid plastic modifications of tumor cells. Nevertheless, despite the described limitations, these markers, being key links in the framework of molecular transformations of cells and functional-population transitions, can be used as relevant at the present stage [11,12,13,14].

We should note that, in addition to the functional versatility of glioblastomas in relation to definitive cell populations, the emergence of two varieties of glioma stem cells (GSCs) is also observed. This cambial pool includes proneuronal GSCs (PGSCs) and mesenchymal GSCs (MesGSCs). The importance of isolating this pool of cells lies not only in the need to correctly build the hierarchy of cellular components in the structure of the diffuse gliomas’ development, but also in the properties of the GSCs themselves [15,16]. This type of cell population has a key part in the occurrence and pathogenesis of tumors, in the formation of chemotherapy and radiation therapy resistance, and in the appearance of tumor relapses [17,18].

In general, the properties of diffuse gliomas’ intratumoral heterogeneity, as expressed in the features of the neoplasms’ cell-population composition and interactions between different populations, seem to be extremely important for the development of neuro-oncology. Nevertheless, many issues of diffuse gliomas’ intratumoral heterogeneity remain poorly understood, and a number of fundamental problems that open the way to the creation of fundamentally new diagnostic and therapeutic approaches, including the issues of dynamic changes in the quantitative and qualitative composition of populations with the glioma progression, have not yet been considered. It is also extremely relevant to compare these dynamics between different types of diffuse gliomas. In this regard, in the framework of the study, we carried out a comparative dynamic analysis of the different cell populations’ content in diffuse gliomas of different molecular profiles and different grades, also considering the functional properties of cell populations, molecular characteristics that affect population dynamics, and the relationship of these parameters with patients’ survival.

## 2. Materials and Methods

### 2.1. General Characteristics

The scientific work was retrospective and randomized; the material of the study was samples of neoplasms obtained during neurosurgical interventions at the Burdenko Neurosurgical Institute and the Brain and Neurotechnology Center. The study included 52 patients with astrocytoma, grade 2, IDH-mut; 58 patients with astrocytoma, grade 3, IDH-mut; 52 patients with astrocytoma, grade 4, IDH-mut; 50 patients with oligodendroglioma, grade 2, IDH-mut and 1p/19q co-deleted; 58 patients with oligodendroglioma, grade 3, IDH-mut and 1p/19q co-deleted; and 54 patients with glioblastoma, IDH-wt. All patients underwent surgery in 2016–2020. The summary patient characteristics with each nosology are given in Table 1.

In the postoperative period, all patients were observed at the oncological consultation at the Burdenko Neurosurgical Institute and the Brain and Neurotechnology Center, and adjuvant therapy was recommended to them. According to the follow-up study, radiation therapy and chemotherapy in regimens selected in accordance with modern international clinical guidelines were performed in all patients included in the study. All patients were regularly followed up at the Burdenko Neurosurgery Institute and the Brain and Neurotechnology Center and subjected to regular periodic medical examinations at least four times a year, with imaging control, using magnetic resonance imaging, as well as positron emission computed tomography, if necessary, in accordance with international recommendations [19,20]. Joining the experimental cohort for the patient became possible in the case of primary surgery for this neoplasm; lack of previous treatment, including chemotherapy and radiation therapy; age over 18 years; and the presence of a single neoplasm at the time of surgery.

As regulatory documents for the organization of the study, the provisions of the Declaration of Helsinki were applied and strictly observed. Permission to conduct the study was obtained from the local ethics committee of the Brain and Neurotechnology Center.

### 2.2. Histopathological Diagnosis

Tumor fragments were placed in pathological blocks with the subsequent formalin fixation (Sigma-Aldrich, St. Louis, MO, USA), and then they were passed through alcohols to dehydrate and degrease tissues, after which the material was impregnated with paraffin. Next, sections that were 3 μm thick were made. The process of paraffin removal was carried out, sequentially passing the material through xylenes and different alcohol concentrations. After that, the slides were washed with water and immersed in hematoxylin for staining cell nuclei for 5 min (Mayer’s hematoxylin, Sigma-Aldrich, St. Louis, MO, USA). Next, the slides were placed in water to remove excess dye. After that, they were cleaned with water and stained with hematoxylin and eosin solution (Sigma-Aldrich, St. Louis, MO, USA). After carbol-xylene solution clarification, a coverslip was placed over them.

Further slides were studied with the participation of three experienced pathologists with proven competencies in the field of neuropathology who, based on pathohistological, genetic (IDH1 and IDH2 genes mutational status, 1p/19q cooperative deletion), and, if necessary, immunohistochemical parameters, made a diagnosis according to the WHO classification criteria [5].

### 2.3. Flow Cytometry

Glioma tissue for flow cytometry was collected during surgical interventions by excision of a small fragment by a neurosurgeon from the area of tumor growth assumed according to intraoperative data and data from different methods of preoperative and intraoperative visualization. Further, the material was subjected to pathohistological control with the manufacture and study of its pathohistological and molecular properties according to the method described above. If the sample contained exclusively or predominantly tumor cells, it was further subjected to flow cytometry.

For this, sample tissue was digested in RPMI and Accumax solution buffer, mixed in 1:1 relation (Thermo Fisher Scientific, Waltham, MA, USA). The mesh of 70 μm was applied to clean tissue with 30% Percoll’s reagent help (Thermo Fisher Scientific, Waltham, MA, USA). After centrifugation—15 min, 1500 G—erythrocytes were lysed, PBS washing was performed, and the buffer–antibodies solution was applied. CD16 and CD32 antibodies were used to inhibit endogenous Fc fragments—15 min reaction, 1:100 dilution (Sigma-Aldrich, St. Louis, MO, USA). Then washing was carried out, and an antibody cocktail was applied that contained antibodies specific to NPC-like cells markers CD24 (Sigma-Aldrich, St. Louis, MO, USA), DCX (Sigma-Aldrich, St. Louis, MO, USA), DLL3 (Sigma-Aldrich, St. Louis, MO, USA), and Sox11 (Sigma-Aldrich, USA); specific AC-like cells markers CST3 (Sigma-Aldrich, St. Louis, MO, USA), HOPX (Sigma-Aldrich, St. Louis, MO, USA), S100B (Sigma-Aldrich, St. Louis, MO, USA), and MLC1 (Sigma-Aldrich, St. Louis, MO, USA); specific OPC-like cells markers PLP1 (Sigma-Aldrich, St. Louis, MO, USA), OLIG1 (Sigma-Aldrich, St. Louis, MO, USA), OMG (Sigma-Aldrich, St. Louis, MO, USA), and PLLP (Sigma-Aldrich, St. Louis, MO, USA); specific MES1 cells markers CHI3L1 (Sigma-Aldrich, St. Louis, MO, USA), ANXA1 (Sigma-Aldrich, St. Louis, MO, USA), ANXA2 (Sigma-Aldrich, St. Louis, MO, USA), and CD44 (Sigma-Aldrich, St. Louis, MO, USA); specific MES2 cells markers DDIT3 (Sigma-Aldrich, St. Louis, MO, USA), ENO2 (Sigma-Aldrich, St. Louis, MO, USA), LDHA (Sigma-Aldrich, St. Louis, MO, USA), and HILPDA (Sigma-Aldrich, St. Louis, MO, USA); specific PGSC cells markers Sox2 and CD133 (Thermo Fisher Scientific, Waltham, MA, USA); and specific MesGSC cells markers CD109 (Thermo Fisher Scientific, Waltham, MA, USA) and WT1 (Thermo Fisher Scientific, Waltham, MA, USA). The process of incubation lasted for one hour. The panel of markers for cell sorting was selected according to the data of single-cell profiling of glioma cell populations in previous studies [11,12]. Absolute cell counts were performed by using a NovoCyte volumetric flow cytometer (ACEA Biosciences, San Diego, CA, USA). Gating was performed by using NovoExpress 1.2.1 software (ACEA Biosciences, San Diego, CA, USA). For the control, the fluorescence minus one with single-channel, unstained experimental variants was deployed. NPC-like cells (CD24+/DCX+/DLL3+/Sox11+/other markers-), AC-like cells (CST3/HOPX/S100B/MLC1/other markers-), OPC-like cells (PLP1+/OLIG1+/OMG+/PLLP+), MES1 cells (CHI3L1+/ANXA1+/ANXA2+/CD44+), MES2 cells (DDIT3+/ENO2+/LDHA+/HILPDA+), PGSC cells (CD133+/Sox2+/CD109-/WT1-), and MesGSC cells (CD109+/ WT1+/CD133-/Sox2-) were identified and separately counted.

### 2.4. Multiparametric Fluorescent In Situ Hybridization

The technique was similar to previous work [14]. Briefly, frozen samples after paraformaldehyde solution (Thermo Fisher Scientific, Waltham, MA, USA) heating were placed in PBS solution. Dehydration was performed with different alcohol concentrations. Dried samples were pretreated with protease IV (Advanced Cell Diagnostics, Bio-Techne, Minneapolis, MN, USA). Preliminary RNA probes were warmed up at 40 °C and diluted to 1:50. Then RNA probes for DLL3 messenger RNAs as specific markers of NPC-like cells, EGFR messenger RNAs as specific markers of AC-like cells, PDGFRA messenger RNAs as specific markers of OPC-like cells, CHI3L1 messenger RNAs as specific markers of MES1 cells, DDIT3 messenger RNAs as specific markers of MES2 cells, CD133 messenger RNAs as specific markers of PGSC cells, and CD109 messenger RNAs as specific markers of MesGSC were used (Thermo Fisher Scientific, Waltham, MA, USA). DLL3 was marked blue, EGFR was red, PDGFRA was green, CHI3L1 was orange, DDIT3 was pink, CD133 was lime, and CD109 was murrey. After being washed with PBS, the samples were humidified in a special chamber, after which they were covered by probes solution in the amount of 50 μL.

After a two-hour incubation, samples were washed with 1x buffer and treated with amplification solution AMP1 (Thermo Fisher Scientific, Waltham, MA, USA). The durability of treatment was half an hour. Then washed samples were placed in amplification solution AMP2 (Thermo Fisher Scientific, Waltham, MA, USA). This solution was applied for 15 min. After this, washed samples were processed by amplification solution AMP3 (Thermo Fisher Scientific, Waltham, MA, USA); after 20 min of incubation, they were washed. Then samples were covered by amplification solution AMP4 (Thermo Fisher Scientific, Waltham, MA, USA), and the period of incubation was fifteen minutes. Washed samples were stained with DAPI (Thermo Fisher Scientific, Waltham, MA, USA) for 30 s. Then samples were covered by 10 μL of Mowiol DABCO aqueous mounting medium (Thermo Fisher Scientific, Waltham, MA, USA) and closed with a coverslip. The samples were stored for 12 h at 4 °C.

Then confocal microscopy was used to quantify signals with a help of ImageJ (NIH, Bethesda, MD, USA) and Imaris 9.4 (Bitplane, Zürich, Switzerland) licensed software.

### 2.5. Immunostaining

Tissue and cell culture samples obtained at different stages of the study were fixed with formalin and embedded in paraffin. Next, sections with a 30 μm thickness were formed, which were deparaffinized with xylene for 30 min and rehydrated in ethanol solutions with a successively increasing concentration from 70% to 100%, with a change every 5 min. After preliminary preparation, the samples were incubated, depending on the stage of the study, with rabbit primary monoclonal antibodies to DLL3, rabbit primary monoclonal antibodies to EGFR, rabbit primary monoclonal antibodies to CHI3L1, rabbit primary monoclonal antibodies to CD133, rabbit primary monoclonal antibodies to CD109, rat primary monoclonal antibodies to PDGFRA, rat primary monoclonal antibodies to DDIT3 (Sigma-Aldrich, USA), and rat primary monoclonal antibodies to HIF-1α (Sigma-Aldrich, St. Louis, MO, USA). The first five antibodies were used at a dilution of 1:50, and the remaining antibodies at a dilution of 1:100; the duration of the incubation process was 24 h, at 4 °C. The mouse secondary antibodies were conjugated with the blue dye for DLL3, red color for EGFR, green color for PDGFRA, orange color for CHI3L1, pink color for DDIT3, lime color for CD133, and dark red color for CD109. Two 15 min washes with PBS were performed after each staining period. To stain the nuclei, the samples were incubated with DAPI dye (Thermo Fisher Scientific, Waltham, MA, USA) at a dilution of 1:10,000 in PBS buffer for 1 h. At the next stage, Vaseline strips 4 mm high were prepared for microscopy on glass slides, followed by the addition of the mounting medium Fluoromount-G (Thermo Fisher Scientific, Waltham, MA, USA), into which the washed samples were placed. A unique glass lid was then attached on top; the edges were sealed with clear varnish.

### 2.6. Confocal Microscopy

Laser scanning confocal microscopy was applied by using Leica STELLARIS (Leica, Wetzlar, Germany). A standard layer thickness of 30 μm was used for all cases, and shooting was carried out with a step of 4.5 μm between layers. ImageJ and Imaris software were applied for image reconstruction. Next, threshold values for the volume of fluorescent labels, ellipticity, and signal quality were set for better differentiation of individual neoplasm cells. Images were then quantified by using ImageJ and Imaris to determine the percentage of cells with positive expression of specific markers.

### 2.7. Real-Time Quantitative Polymerase Chain Reaction

Complete RNA extraction from cell samples of all clusters was performed by using the TRIzol reagent (Thermo Fisher Scientific, Waltham, MA, USA) and then treated with a reverse-transcription kit (Thermo Fisher Scientific, Waltham, MA, USA). cDNA was quantified by quantitative PCR, using a QuantStudio 7 Flex Real-Time PCR system (Thermo Fisher Scientific, Waltham, MA, USA). The composition of the forward and reverse primers is shown in Table 2. A set of genes for assessing the metabolic status of cells was created on the basis of large studies of key metabolic pathways in different cancer types [21]. The results were calculated by the 2-ΔΔCT method, standardized for the U6 gene.

### 2.8. Cell Cultures

Cultural studies were realized similarly to previously performed work [14]. Briefly, cell cultures were created by using C57BL/6m mice at the age of 5–6 weeks. Hemisphere slice culture simulated cultural tumor model. Six-well Millicell Cell Culture Insert plate (Merck, Rahway, NJ, USA), Dulbecco’s modified Eagle medium, and serum-free primary NSC medium, (Thermo Fisher Scientific, Waltham, MA, USA) were used to create a cell culture. They were incubated in hypoxic conditions for two or three days. Flow cytometry was applied for cell sorting. After that, tumor cells were placed into classical and hypoxic conditions for a week.

### 2.9. Statistical Analysis

For statistical processing, SPSS Statistics 26.0 software (IBM, Armonk, NY, USA) was applied. To find out distribution features, the Shapiro–Wilk test was applied. Within the intergroup comparisons for abnormal distribution, the Mann–Whitney U test was used, while for normal distribution, Student’s *t*-test was deployed. Statistical significance of the studied factors’ effect on survival was evaluated with usage of Cox proportional hazards and Kaplan–Meier curves. Differences were considered significant at *p* < 0.05.

## 3. Results

### 3.1. Cell Populations Dynamics in Diffuse Gliomas, IDH-Mut

Within the sample cohort, we first assessed the content of different cell populations in astrocytomas, IDH-mut, grade 2, and in oligodendrogliomas, IDH-mut, grade 2, by flow cytometry. The results obtained, in general, corresponded to the results of earlier studies, according to which there were no significant differences at the cell-population level between the two considered nosological units (*p* = 0.842, Figure 1a). The main cell mass consisted of AC-like and OPC-like cell populations in approximately equal proportions; PGSCs acted as the cambial pool (Figure 1b). The obtained data were confirmed by mrFISH and immunofluorescence on the patient material (Figure 1c–f).

Next, we studied the cell-population composition in astrocytomas, IDH-mut, grade 3, and in oligodendrogliomas, IDH-mut, grade 3, without the necrosis and vascular endothelium proliferation, using flow cytometry, and compared the results with diffuse gliomas, IDH-mut, grade 2, as discussed above (Figure 2a). Although, in general, the qualitative composition of cell populations was preserved, as well as the quantitative ratio of AC-like and OPC-like cell populations, nevertheless, the amount of PGSC significantly increased in diffuse gliomas, IDH-mut, grade 3, compared with diffuse gliomas, IDH- mut, grade 2 (*p* < 0.001, Figure 2b). The data obtained were confirmed by mrFISH and immunofluorescence on patient material (Figure 2c,f). Additionally, PCR revealed a significant difference in the PGSC marker genes’ expression between diffuse gliomas, IDH-mut, grade 2, and diffuse gliomas, IDH-mut, grade 3 (*p* < 0.001, Figure 2d), and the absence of such a difference for AC-like and OPC-like cell populations’ driver genes (*p* = 0.828 and *p* = 0.834, respectively, Figure 2e).

We then turned to the study of the cell-population composition of astrocytomas, IDH-mut, grade 4, and of oligodendrogliomas, IDH-mut, grade 3, with the presence of necrosis and/or vascular endothelium proliferation (Figure 3a). In general, a continuation of the previously outlined trend was revealed with the parity of AC-like and OPC-like cell populations remaining, as well as a further significant increase in the number of PGSCs (*p* = 0.004, Figure 3b). At the same time, with an increase in the neoplasms’ malignancy degree, a fundamentally new cell population began to be detected in a significant amount, previously not found in diffuse IDH-mut gliomas—MES-like cells (Figure 3a). Moreover, the presence of MES2 subtype cells was predominantly determined, and only a small number of MES1 subtype cells were determined (Figure 3a). The data obtained were confirmed by mrFISH and immunofluorescence on patient material (Figure 3c,f–h). In addition, PCR revealed a significant difference in the PGSC marker genes expression between diffuse gliomas, IDH-mut, grade 3, and astrocytomas, IDH-mut, grade 4, as well as oligodendrogliomas, grade 3, with signs of further malignancy oligodendrogliomas, IDH-mut, grade 3, with no signs of malignancy from the previous stage (*p* < 0.001, Figure 3d), and a significant difference between the same groups in terms of MES-like cell population driver genes (*p* < 0.001, Figure 3e).

Thus, we have shown that an increase in the malignancy degree of astrocytomas (IDH-mut) and oligodendrogliomas (IDH-mut) is accompanied not only by an increase in the proportion of stem cells, as indicated by previous works, but also by the appearance of cell populations with mesenchymal molecular properties.

### 3.2. MES-like Cell Population Genesis Identification in Diffuse Gliomas, IDH-Mut

To reveal the mechanisms of the emergence of cells with mesenchymal properties in diffuse gliomas, IDH-mut, we conducted cultural studies, followed by cell-population typing. First, we sorted diffuse glioma cells, IDH-mut, grade 2, grade 3, and grade 4, based on flow cytometry to isolate PGSC, AC-like, and OPC-like cell populations. Next, we performed separate seeding on cell cultures of all varieties of cell tumor populations and placed them in an incubator in which these cultures were in conditions of relatively low oxygen content and high CO_2_ content. As a result, using flow cytometry, it was revealed that only in the culture of OPC-like cells obtained from the astrocytomas, IDH-mut, grade 3 and grade 4, and from oligodendroglioma, IDH-mut, grade 3, but not in the culture of similar neoplasms having grade 2, the appearance of MES-like cells is observed (Figure 4a). Moreover, it is important to note that, as in the patient samples, there was a significant predominance of MES2 subtype cells compared to MES1 subtype cells (Figure 4b). The data obtained were confirmed by mrFISH and immunofluorescence on culture material (Figure 4c,d). Moreover, we validated the considered results on the patients’ material samples. To do this, we identified a group of astrocytomas, IDH-mut, grade 3 and grade 4, and oligodendroglioma, IDH-mut, grade 3 (*n* = 88), with a high level of the HIF-1α hypoxic marker and a group of the same tumor types (*n* = 80) with a low expression of HIF-1α marker (Figure 4e). It was found that, in the group with increased molecular signs of hypoxia, the content of MES2 and, to a lesser degree, MES1 cells was significantly higher compared to the group with low levels of hypoxic marker expression in cells (*p* < 0.001 and *p* = 0.004, respectively).

In order to find out what molecular mechanism may be behind the different ability to perform the mesenchymal transformation of OPC-like cells obtained from diffuse gliomas of different grades, we decided to compare OPC-like cells isolated by using flow cytometry-based sorting from astrocytoma, IDH-mut, grade 3 and grade 4, as well as oligodendrogliomas, IDH-mut, grade 3, with their analogs isolated from the same neoplasms that had grade 2. The subject of comparison was the expression profiles of a large number of metabolic state genes, since we showed above that the modification of precisely metabolic conditions is capable of inducing transformation processes; they were assessed by using PCR. It was found that significant differences between these cell types are observed in the molecular programs of cell plasticity and cell response to hypoxia (Table 3).

Thus, the data reviewed suggest a key role for the OPC-like cell population as the source of the appearance of cells with mesenchymal properties in diffuse gliomas, IDH-mut. At the same time, only those OPC-like cells that demonstrate activation of molecular programs of cell plasticity and cell response to hypoxia, localized mainly in the tissue of astrocytomas, IDH-mut, grade 3 and grade 4, and in the tissue of oligodendrogliomas, IDH-mut, grade 3, can be a source of mesenchymal transformation.

### 3.3. Cell-Population Dynamics in Glioblastomas

At the beginning, we decided to analyze the cell-population composition of diffuse gliomas, which, despite the absence of classical pathohistological signs of glioblastomas—primarily necrosis and vascular endothelial proliferation—were diagnosed as glioblastomas based on the results of molecular typing, including identification of the IDH-wild type molecular status (Figure 5b). Among these tumors, at the first stage, we collected a group of gliomas with low proliferative activity and a Ki-67 labeling index (LI) less than 5%, corresponding to diffuse glioma, grade 2; that is, this group can be considered an early stage of development of glioblastoma with low proliferative activity and before the appearance of its traditional pathohistological characteristics [22,23]. As a result, using flow cytometry, a significantly more variegated cell-population pattern was revealed, which included in the stem component, in addition to PGSC, MesGSCs in a small amount (4.42 ± 2.44% of the total cells number) in 62.96% of cases (*n* = 34), as well as AC-like, OPC-like, and NPC-like definitive cell populations (Figure 5a). The data obtained were confirmed by using mrFISH and immunofluorescence on the patient material (Figure 5c–e).

Next, we took a group of glioblastomas with the absence of classical pathohistological signs of this neoplasm but with increasing proliferative activity and Ki-67 LI of more than 5% but less than 10%, which serves as a threshold value for grade 4 diffuse gliomas [21,22]. Using flow cytometry, we detected a significant increase in the number of MesGSCs compared to the previous group (*p* < 0.001, Figure 6a,b). In addition, MES-like cell populations appeared in the cell composition, including MES1 and MES2 cell subpopulations (Figure 6a,b). The data obtained were confirmed by mrFISH and immunofluorescence on patient material (Figure 6c–e).

Then, in a comparative manner, we studied the cell-population composition in glioblastomas with the presence of classical pathohistological diagnostic criteria. In these glioblastomas, compared to the previous group, a further significant increase in the number of MesGSCs was observed (*p* = 0.008, Figure 7a,b). There was also a significant increase in the number of both MES1 and MES2 cell populations compared with the previous group (*p* < 0.001, Figure 7a,b). Thus, with an increase in the glioblastoma malignancy degree, there was an increase in the quantitative content of both stem and definitive cells with mesenchymal differentiation.

### 3.4. MesGSC and MES-like Cell Population Genesis Identification in Glioblastomas

In order to reveal the mechanisms of the emergence of MesGSC and MES-like cell populations in glioblastomas, we carried out cultural studies, followed by the typing of cell populations. First, we performed flow-cytometry-based sorting of glioblastoma cells with the isolation of PGSC, AC-like, OPC-like, and NPC-like cell populations. Next, we performed separate seeding on cell cultures of all varieties of tumor cell populations and placed them in an incubator in which these cultures were in conditions of relatively low oxygen content and high CO_2_ content. As a result, using flow cytometry, it was revealed that only in the PGSC cell culture was the appearance of the MesGSC cell population observed (Figure 8a). Further incubation of these cell cultures under the same conditions resulted in the appearance of both subtypes of MES-like cells in them (Figure 8b). The data obtained were confirmed by mrFISH and immunofluorescence on culture material (Figure 8c–e). Moreover, we validated the considered results on the patients’ material samples. To do this, we identified a group of tumors (*n* = 28) with a high level of the HIF-1α hypoxic marker and a group (*n* = 26) with a low expression of this marker among the glioblastomas. It was found that, in the group with increased molecular signs of hypoxia, the content of MesGSC was significantly higher compared to the group with low levels of hypoxic marker expression in cells (*p* < 0.001). Moreover, in the group with increased production of the hypoxic marker, a significantly higher content of MES1 and MES2 cell populations was observed than in the group with low values of the hypoxia marker (*p* < 0.001 for both populations; see Figure 8f).

Thus, it can be concluded that PGSCs are the most likely source of MesGSCs, and MesGSCs further give rise to MES-like cells under hypoxic conditions.

### 3.5. Cell Populations Effect on Patient Prognosis

Considering the clinical significance of our results, we assessed the impact of different cell populations on relapse-free survival (RFS) and overall survival (OS) in patients with diffuse gliomas. It was shown that, in patients with diffuse gliomas, IDH-mut, the number of MES-like cells has a strong significant negative effect on both the RFS (*p* = 0.002) and OS (*p* < 0.001): the number of MES-like cells negatively correlated with the RFS and OS. Moreover, it turned out that this parameter also negatively affects the RFS (*p* = 0.032) and OS (*p* = 0.024) of patients with glioblastomas. It was also found that the degree of activation of the previously described programs of plasticity and cell response to hypoxia negatively affects the RFS (*p* < 0.001) and OS (*p* < 0.001) in patients with diffuse gliomas, IDH-mut. In addition, it was found that the number of MesGSCs was also significantly negatively correlated with the prognosis of the RFS (*p* < 0.001) and OS (*p* < 0.001) of patients with glioblastoma (Figure 9).

## 4. Discussion

Despite the novelty of the approach of cell populations’ differentiation within the neoplasms according to the molecular principle, this tool is increasingly included in fundamental medicine. The advantages of this method are quite wide since it allows us to consider the finer properties of tumors in the fullness of their individual evolutionary variability. A deeper understanding of molecular characteristics, including diffuse gliomas, leads to the creation of more effective methods for diagnosing and treating patients [24,25,26,27].

An important role in a more complete disclosure of the properties of both neoplasms in general and tumor cell populations in particular is played not only by the description of intratumoral heterogeneity as a static molecular phenomenon, but by the coordination of the detected molecular patterns with the functional parameters of different cell populations. To solve this problem in relation to diffuse gliomas, in this work, we considered the cell-population composition of different types of diffuse gliomas at different malignancy stages, making appropriate internosological comparisons. After careful, meticulous tracing of diffuse glioma cell populations, not only previously identified patterns were confirmed, but a number of new interesting phenomena were also established. In particular, perhaps the key source of malignancy in diffuse gliomas carrying IDH gene mutations is not only an increase in the number of cells with stem properties, as was established earlier, but also a gradual adjustment of a part of the cell mass to an increase in tumor proliferative activity, which entails an inevitable increase in the need for oxygen, coupled with other nutrients, forming a hypoxic environment for neoplasm cells. Switching on the mechanisms of hypoxic adaptation naturally leads to the appearance of cells with mesenchymal properties, and the predominant position is occupied by cells of the MES2 subpopulation, characterized by their high hypoxic resistance.

Furthermore, we went further by showing that, from a cell-population standpoint, OPC-like neoplasm cells serve as a substrate for the formation of mesenchymal-like cell populations in diffuse astrocytomas and oligodendrogliomas. According to previous works, this cell population is generally highly plastic and has, on average, the highest proliferative potential among all glioma cell populations [11,12,28]. In our work, it was shown that the implementation of this functional plasticity in the form of mesenchymal transformation occurs through the activation of molecular programs of cellular plasticity and cell response to hypoxia. At the same time, it should not be forgotten that the considered properties primarily concerned OPC-like glioblastoma cells, which a priori have a slightly different mutational and, in general, molecular background [26,27]. In this regard, despite the obvious similarity in terms of functionality, there are also significant differences in the transition to the most malignant gliomas’ form.

Within the framework of glioblastoma, the path of mesenchymal transformation is more complex, passing through the stem-cell link. Here, we have shown that the direct source of definitive mesenchymal cell subpopulations is MesGSCs, which are not detected in astrocytomas and oligodendrogliomas. Moreover, we have shown that this GSC type is formed from PGSC due to their transformation under the influence of hypoxic factors. In general, the considered differences in the fundamental parameters of diffuse glioma carcinogenesis indicate a deeper and more qualitative adaptation of diffuse glioma cells without IDH mutation to unfavorable environmental conditions, which form the foundation of more malignant potential of this neoplasm type.

It should be noted that, as we already pointed out in the Introduction, there are some methodological limitations in our study. In particular, the markers used to identify cell populations may not always adequately reflect changes in the plastic status of cells. Although the current studies suggest the fundamental possibility of using these markers, nevertheless, an important further direction of research may be the verification of our results using other methodological approaches. This will not only confirm or refine the results of the present work, but also let us obtain a more comprehensive picture of the cell-population tumor pathogenesis.

## 5. Conclusions

In this work, we constructed a functional projection of the molecular properties of intratumoral heterogeneity by tracing cell populations of different diffuse gliomas in a comparative manner. This projection made it possible for the first time to identify mesenchymal-like cellular transformation in astrocytomas and oligodendrogliomas as possibly the key mechanism of malignancy. In the case of glioblastoma, it was also shown for the first time that mesenchymal transformation is of a deeper nature, passing through the stem-cell link. The involvement of cells with stem properties makes it possible to use a greater plastic potential contained in these cells that is realized at the pathogenetic level in the form of an increase in malignant potential. Furthermore, a more detailed consideration of the molecular and functional aspects of intratumoral heterogeneity will make it possible to create fundamentally new therapeutic tools.

## Figures and Tables

**Figure 1 cancers-15-00145-f001:**
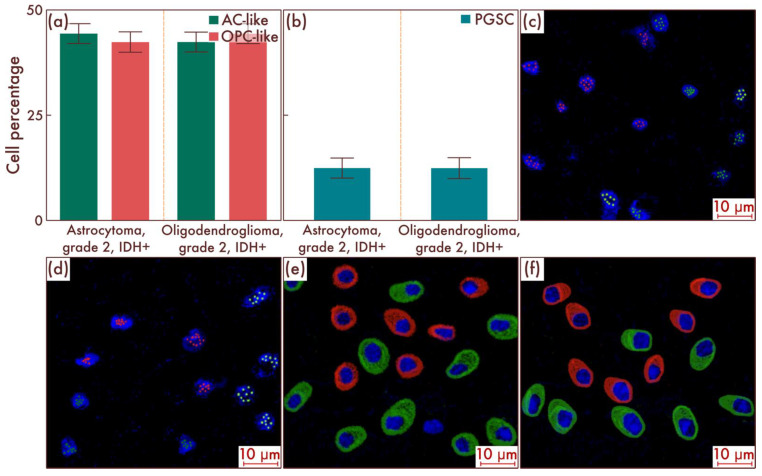
Cell-population composition in astrocytomas, IDH-mut, grade 2, and oligodendrogliomas, IDH-mut, grade 2. (**a**) Ratio of the quantitative content of astrocyte-like cell (AC-like) and oligodendrocyte-progenitor-like cell (OPC-like) populations as a percentage of the total number of tumor cells. (**b**) Quantitative content of proneuronal glioma stem cells (PGSCs) as a percentage of the total number of tumor cells. The results were confirmed on astrocytomas, IDH-mut, grade 2 (**c**), and on oligodendrogliomas, IDH-mut, grade 2 (**d**), by multiparametric fluorescence in situ hybridization and immunofluorescence (**e**,**f**), on which the red marks correspond to EGFR, and green marks to PDGFRA.

**Figure 2 cancers-15-00145-f002:**
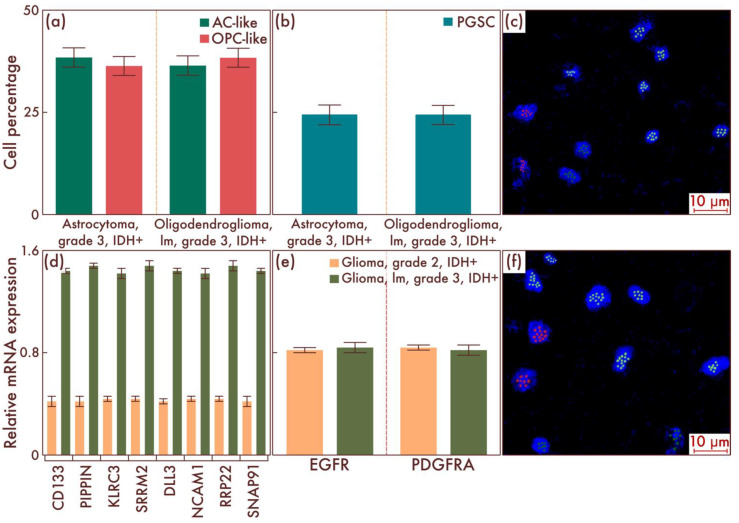
Cell-population composition in astrocytomas, IDH-mutant, grade 3, and in oligodendrogliomas, IDH-mutant, grade 3, with relatively low malignancy (lm). (**a**) Ratio of the quantitative content of astrocyte-like cell (AC-like) and oligodendrocyte-progenitor-like cell (OPC-like) populations as a percentage of the total number of tumor cells. (**b**) Quantitative content of proneuronal glioma stem cells (PGSCs) as a percentage of the total number of tumor cells. A comparison by PCR of astrocytomas, IDH-mut, grade 3 (**c**), and oligodendrogliomas, IDH-mut, grade 3, lm, with astrocytomas, IDH-mut, grade 2, and oligodendrogliomas, IDH-mut, grade 2, in terms of expression activity of definitive (**e**) and stem-cell populations’ (**d**) genes was performed. The results were confirmed on astrocytomas, IDH-mut, grade 3 (**c**), and on oligodendrogliomas, IDH-mut, grade 3, lm (**f**), by multiparametric fluorescence in situ hybridization, on which the red marks correspond to EGFR, and the green marks to PDGFRA.

**Figure 3 cancers-15-00145-f003:**
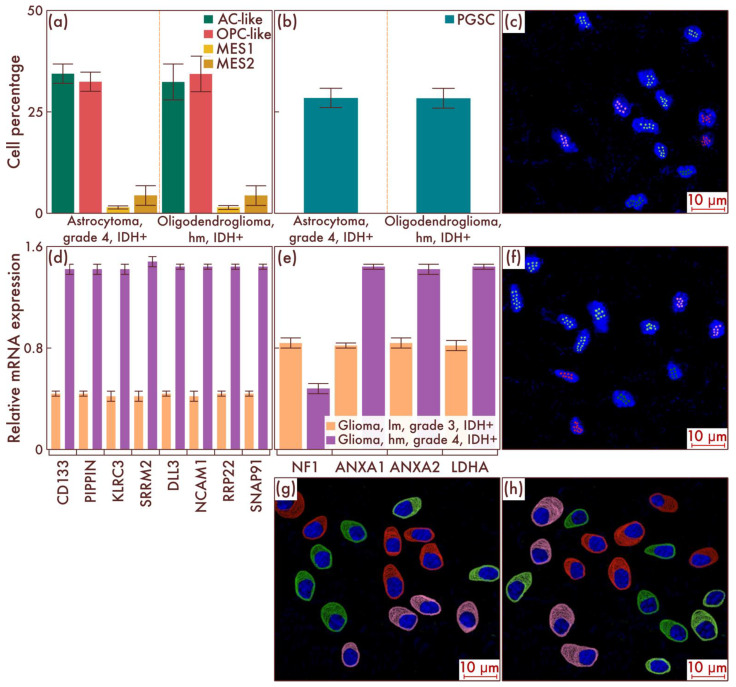
Cell-population composition in astrocytomas, IDH-mutant, grade 4, and in oligodendrogliomas, IDH-mutant, grade 3, with a relatively high malignancy (hm). (**a**) Ratio of the quantitative content of astrocyte-like cells (AC-likes), oligodendrocyte-progenitor-like cells (OPC-like) populations, and mesenchymal-like cell population (MES-like) with MES1 and MES2 subpopulations in percentage of the total number of tumor cells. (**b**) Quantitative content of proneuronal glioma stem cells (PGSCs) as a percentage of the total number of tumor cells. Astrocytomas, IDH-mutant, grade 4, and oligodendrogliomas, IDH-mutant, grade 3, hm, were also compared by PCR with astrocytomas, IDH-mut, grade 3 (**c**), and oligodendrogliomas, IDH-mutant, grade 3, lm, in relation to gene expression activity of mesenchymal definitive cell populations (**e**) and stem cells (**d**). The results were confirmed on astrocytomas, IDH-mutant, grade 4 (**c**), and on oligodendrogliomas, IDH-mutant, grade 3, hm (**f**), by multiparametric fluorescence in situ hybridization and immunofluorescence (**g**,**h**), on which the red marks correspond to EGFR, green marks to PDGFRA, pink marks to DDIT3, and lime marks to CD133.

**Figure 4 cancers-15-00145-f004:**
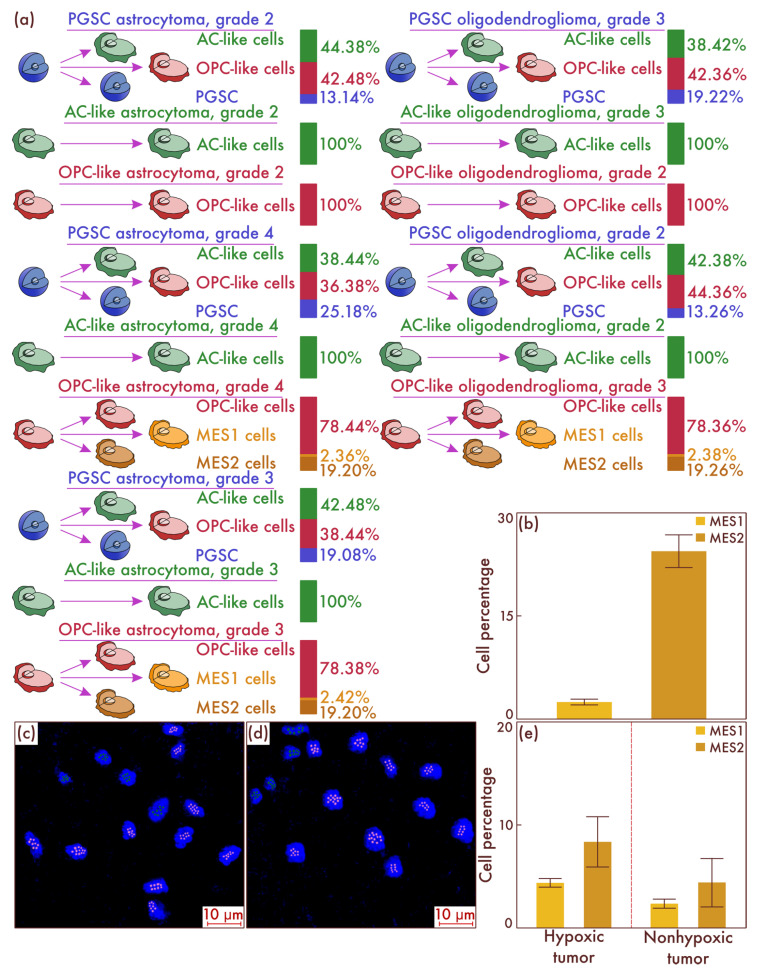
Tracing the cell fate of different tumor cells’ populations isolated in the tissue of astrocytomas and oligodendrogliomas of different grades placed in hypoxic conditions (**a**). Histogenetic transformation pathways of astrocyte-like cells (AC-likes), oligodendrocyte-progenitor-like cells (OPC-likes), and proneuronal glioma stem cells (PGSCs) are presented. The quantitative ratio of mesenchymal-like cell population MES1 and MES2 in the cell culture formed as a result of seeding PGSC from astrocytomas, IDH-mut, grade 3 and grade 4, and from oligodendroglioma, IDH-mut, grade 3, as a percentage of the total cell mass is also shown (**b**). The results were confirmed in cultures of astrocytomas, IDH-mutant, grade 4 (**c**), and in oligodendrogliomas, IDH-mutant, grade 3 (**d**), by multiparametric fluorescence in situ hybridization, on which green marks correspond to PDGFRA, and pink marks correspond to DDIT3. The results of a comparative study (**e**) of the mesenchymal-cell-population composition of glioblastomas both with and without the HIF-1α hypoxia marker increase are presented in the material of patients’ neoplasms.

**Figure 5 cancers-15-00145-f005:**
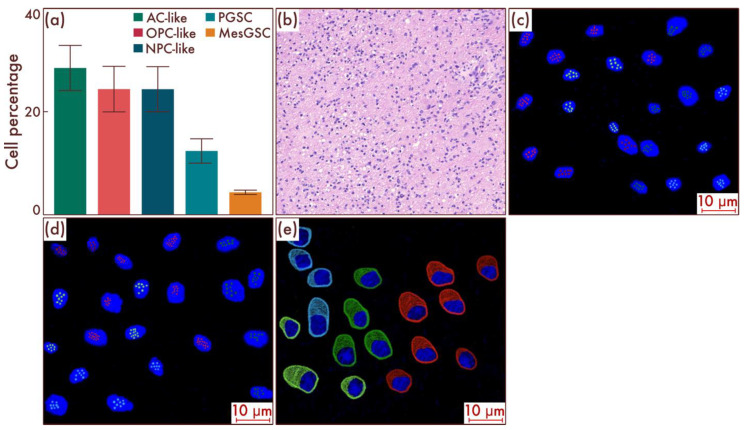
Cell-population profiling in glioblastomas, grade 4, without classical pathohistological criteria in the form of necrosis and vascular proliferation (**a**). (**b**) Ratio of the quantitative content of astrocyte-like cells (AC-likes), oligodendrocyte-progenitor-like cells (OPC-likes), neural progenitor-like cell (NPC-like) populations, proneuronal glioma stem cells (PGSCs), and mesenchymal glioma stem cells (MesGSCs) as a percentage of the total tumor cell mass. The results were confirmed on glioblastomas (**c**,**d**), using multiparametric fluorescence in situ hybridization and immunofluorescence (**e**), on which the red marks correspond to EGFR, green marks to PDGFRA, blue marks to DLL3, pink marks to DDIT3, lime marks to CD133, and dark red marks to CD109.

**Figure 6 cancers-15-00145-f006:**
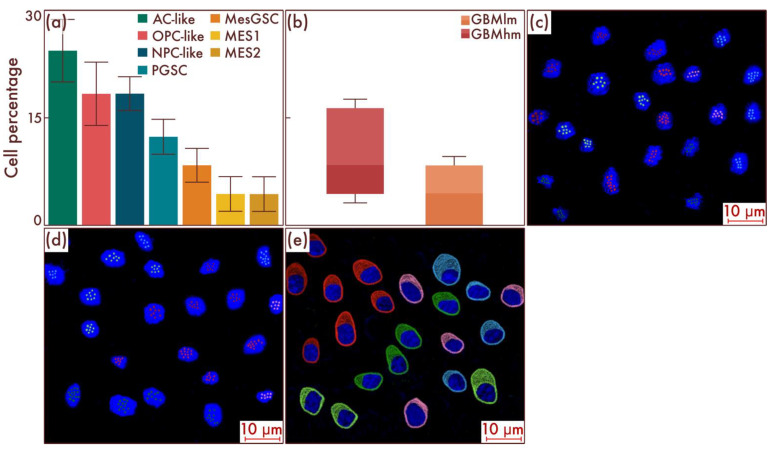
Cell-population profiling in glioblastomas with increasing proliferative activity and Ki-67 LI 5–10%. (**a**) Ratio of the quantitative content of astrocyte-like cells (AC-likes), oligodendrocyte-progenitor-like cells (OPC-likes), neural progenitor-like cell (NPC-like) populations, mesenchymal-like cell population (MES-like) with MES1 and MES2 subpopulations, proneuronal glioma stem cells (PGSCs), and mesenchymal glioma stem cells (MesGSCs) as a percentage of the total tumor cell mass. Moreover, the ratio of the quantitative content of MesGSCs in glioblastomas with increasing proliferative activity and Ki-67 LI 5—10% and glioblastomas, grade 4, without classical pathohistological criteria in the form of necrosis and vascular proliferation was shown (**b**). The results were confirmed on glioblastomas (**c**,**d**), using multiparametric fluorescence in situ hybridization and immunofluorescence (**e**), on which the red marks correspond to EGFR, green marks to PDGFRA, blue marks to DLL3, pink marks to DDIT3, lime marks to CD133, dark red marks to CD109, and pink marks to DDIT3.

**Figure 7 cancers-15-00145-f007:**
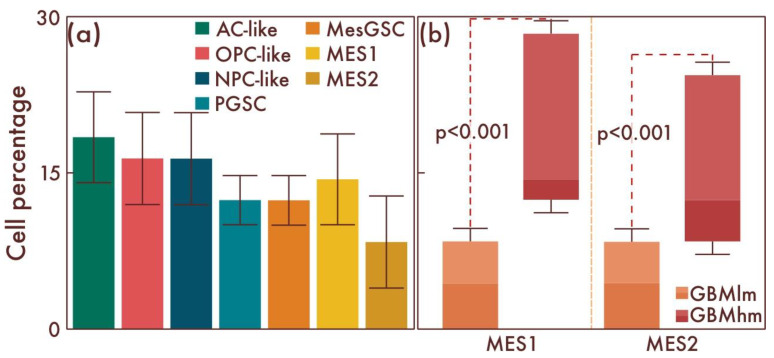
Cell-population profiling in glioblastomas with the presence of classical pathohistological diagnostic criteria. (**a**) Ratio of the quantitative content of astrocyte-like cells (AC-likes), oligodendrocyte-progenitor-like cells (OPC-likes), neural progenitor-like cell (NPC-like) populations, mesenchymal-like cell population (MES-like) with MES1 and MES2 subpopulations, proneuronal glioma stem cells (PGSCs), and mesenchymal glioma stem cells (MesGSCs) as a percentage of the total tumor cell mass. The ratio (**b**) of the quantitative content of MES1 and MES2 cell populations in glioblastomas with the presence of classical pathohistological diagnostic criteria and glioblastomas with increasing proliferative activity and Ki-67 LI 5—10% was shown.

**Figure 8 cancers-15-00145-f008:**
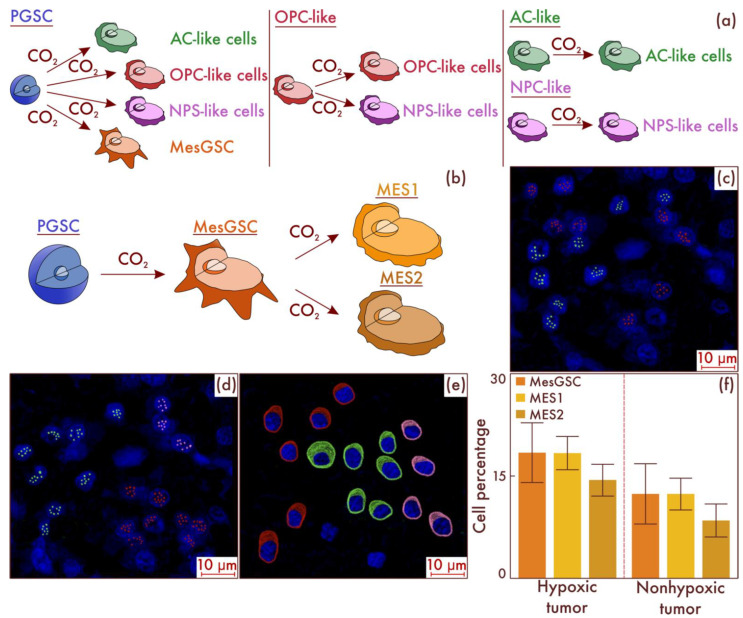
Fate tracing of different glioblastoma cell populations in a hypoxic environment (**a**). Histogenetic transformation pathways of proneuronal glioma stem cell (PGSC), astrocyte-like cell (AC-like), oligodendrocyte-progenitor-like cell (OPC-like), and neural progenitor-like cell (NPC-like) populations are presented. In addition, (**b**) a further path of evolution under hypoxic conditions of mesenchymal glioma stem cells (MesGSCs) derived from PGSC with the formation of definitive mesenchymal populations was demonstrated. The results were confirmed in glioblastoma (**c**,**d**) patient samples, using multiparametric fluorescence in situ hybridization and immunofluorescence (**e**), on which lime marks correspond to CD133, dark red marks to CD109, and pink marks to DDIT3. The results of a comparative study (**f**) of the mesenchymal-cell-population composition of glioblastomas with and without the HIF-1α hypoxia marker increase are presented in the material of patients’ neoplasms.

**Figure 9 cancers-15-00145-f009:**
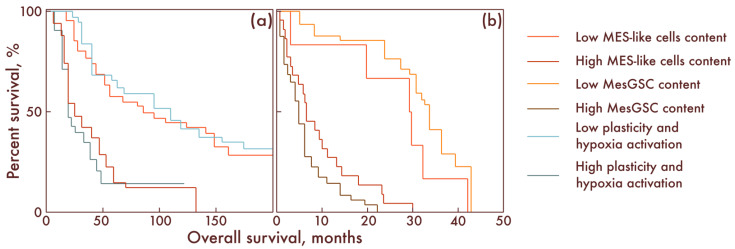
Kaplan–Meier curves demonstrating overall survival of patients with low and high MES-like cells and low and high activation programs of plasticity and cell response to hypoxia in diffuse gliomas, IDH-mut (**a**), as well as with low and a high content of MES-like cells and MesGSC in glioblastomas (**b**). The overall survival is shown along the abscissa in months; along the ordinate axis, the percent survival is shown.

**Table 1 cancers-15-00145-t001:** The most important patient characteristics for the respective glioma types.

	Astrocytoma, Grade 2, IDH-mut	Astrocytoma, Grade 3, IDH-mut	Astrocytoma, Grade 4, IDH-mut	Oligodendroglioma, Grade 2, IDH-mut	Oligodendroglioma, Grade 3, IDH-mut	Glioblastoma, IDH-wt
Mean age (years)	34.8 ± 4.2	38.2 ± 8.2	48.2 ± 4.8	38.0 ± 2.8	48.8 ± 4.2	68.4 ± 8.6
Men and women ratio (percentage)	54.58% to 45.42%	61.54% to 38.46%	54.84% to 45.16%	52.00% to 48.00%	54.18% to 45.82%	52.86% to 47.14%
Most frequent localization	Frontal lobe	Frontal lobe	Frontal lobe	Frontal lobe	Frontal lobe	Temporal lobe
Contrast accumulation (percentage)	2.84%	62.07%	94.24%	12%	82.76%	98.14%
Disease-free survival (months)	58.4	22.8	14.2	58.8	28.4	5.8
Overall survival (months)	136.8	112.2	28.8	144.2	94.2	16.4

**Table 2 cancers-15-00145-t002:** Complete list of genes analyzed by polymerase chain reaction with their respective forward and reverse primers.

Genes	Forward	Reverse
CD133	AGA-TCC-CCA-CGC-GAG- GGG	CGG-CAC-CTG-GCC- CCG-CCC
PIPPIN	CTG-AGG-GCA-GTT-ATC-ATA	GCG-CGA-CCC-TCA-CAT-CAC
KLRC3	GTG-CTG-ACG-TGC-AGC-CCA	GGC-GGG-GTG-CGA-GTG-GCG
SRRM2	TTG-GAG-CCC-GTT-GCG-GCC-CCT-GAG	CGA-GGA-GGC-GTC-GGC-GTC-GGC-TGA
DLL3	GAG-GCG-AGG-CGG-CGG-CCC	GAC-TCG-GGA-GCT-CGA-GCA
NCAM1	CAA-GAC-CTC-TCC-CCC-TCG	TGA-GAG-GGT-GAG-GGA-GCC
RRP22	GCG-AGC-CGG-GTG-GGG-GAG	GCC-CCG-GCG-CGA-GGC-CAA
SNAP91	GGA-GAA-CTG-CCC-AGG-CCG-CCG-AGC	CTC-CGC-GCG-CAC-GCG-CGC-TCG-ACG
EGFR	CCT-CGC-CTC-CCT-TCC-CCC	CCG-TTC-CTG-TCG-CGC-GCG
PDGFRA	TGA-CAA-CGG-CGT-TCG-CAG	CTG-TAC-TCG-GGT-TCG-CGA
NF1	TTG-GTG-AGG-GGG-GAG-GGC	CTC-GGC-TTC-ACT-CCT-CCC
ANXA1	CAA-CAT-GTG-CCG-CCG-AGG-TGG-CCT	TGC-TCC-TGG-ACC-CTT-GCC-CCC-GGT
ANXA2	CCT-CGG-GCT-TCT-CTT-CGG	GCT-GCC-CGT-GTC-GGG-GGC
LDHA	CCC-TCC-CCC-ATT-CCG-CAA	CGA-AAA-TAG-GCC-CCC-GGC
SLC2A1	TTG-AAG-GAA-TAA-AAT-ATC	TCT-TTA-AAT-ATT-TGG-AAT
SLC2A2	CTG-GGA-GGT-GAA-GTA-TAT-GAC-GAT	CCA-AGG-TTC-ATG-TGT-GTT-TCT-GAT
SLC2A3	TGG-GGT-GTG-AGG-CTG-CAC	CAA-CTA-GTC-TGC-TGT-TCA
SLC2A4	TTC-ATA-TTC-GTA-TAG-GGT	TTA-GGT-AAG-TTT-TTG-GGA
ADPGK	GGC-CGT-TGG-GTG-CCA-CAG	TGG-TGT-TTG-TTG-TGA-CCA
GCK	TTA-GGA-ATT-AGA-GTA-GTT-GGT-AAC	CAC-TTG-TGC-ATT-TTG-GGG-CCA-AAG
PFKFB2	GTG-TGG-AGA-GAT-TTT-AGA	TGT-TTG-CCT-AGG-GTG-GAA
PGK1	GGG-AGG-GAT-TAT-TTA-GAA	GCA-TTG-AGA-GAA-AGG-CTG
LDHA	TAG-AAG-GAA-ATT-TGG-AAG	TTT-CTT-GGG-AGA-CCA-GGA
SLC16A3	CGG-GCA-CGG-GGC-CAT-GTA-CAA-CGG	TGC-CGA-CGC-CCC-GGG-GCA-GCG-GCA
SLC16A5	CGT-CCA-GCG-CAA-CCT-GTC	CGC-CGG-GGT-GAG-CGG-CCT
SLC16A10	CTG-CGG-CGC-CTG-GAG-GCT	CGG-CCT-AAT-CCT-GAC-ATC
BCAT1	GCG-TCG-AGC-TGC-GAT-GCC	GAG-ATG-ATG-GAA-GAG-CAG
GBE1	GTC-AAG-CAC-TGA-ATG-AGT-GCA-GAG	TAG-GTG-GGA-TGT-ATA-GGG-AGC-TTA
GFPT1	GAG-GCC-TGG-CAA-AGA-GTT	GAG-GAG-GCA-GAT-GAG-CTC
G6PD	CAG-TGC-AGA-GTG-GGA-AAT	TGT-GTC-TGG-GGC-ATA-GAT
FADS2	GCA-CGT-GGG-AGA-AGG-GAG	GAA-CAA-AAA-TGT-GTT-TTA
PDHA1	CCA-GGT-AGA-GGA-GCT-AGG-TAA-GCT	TCA-TTG-GTA-GAA-AAG-AAG-AAG-ATA
HSD17B10	GGA-TTG-AGG-CCT-TGG-AAG	GAA-TTA-GAA-GAA-TAC-GTG
Pref-1	CAT-GGC-TGG-GAA-GGG-AAT	GCC-CAT-TAA-GGC-AGA-AAC
ALDH	TAA-ACT-TCA-GCA-ATG-GTG	TAT-CCT-TAC-TTG-GAA-TCC
HIF1A	TTC-TTG-TTG-AAG-TTG-GGA	CTG-CTT-GCT-TAA-CTT-TTC
NANOG	GTA-GCA-TAT-TTA-AAT-CCA	GAG-TAG-TAA-GTA-GAG-GGA

**Table 3 cancers-15-00145-t003:** Genes with statistically significant different levels of expression in OPC-like cells derived from astrocytoma, IDH-mut, grade 3 and grade 4, as well as oligodendrogliomas, IDH-mut, grade 3, and similar cells isolated from the same neoplasms that had grade 2.

Genes	Diffuse Gliomas, IDH-Mut, Grades 3 and 4	Diffuse Gliomas, IDH-Mut, Grade 2
BCAT1	2.84 ± 0.48	1.42 ± 0.24
GBE1	2.16 ± 0.42	1.52 ± 0.18
GFPT1	1.92 ± 0.36	1.46 ± 0.22
G6PD	2.64 ± 0.38	2.12 ± 0.16
FADS2	1.90 ± 0.84	0.78 ± 0.14
PDHA1	2.24 ± 0.82	0.92 ± 0.24
HSD17B10	2.44 ± 0.44	1.86 ± 0.84
Pref-1	2.34 ± 0.36	1.52 ± 0.82
ALDH	2.29 ± 0.94	0.74 ± 0.18
HIF1A	2.28 ± 0.48	1.05 ± 0.14
NANOG	2.32 ± 0.34	0.48 ± 0.16

## Data Availability

The datasets used and/or analyzed during the current study are available from the corresponding author upon reasonable request.

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
