# Peer review of "Cell-Population Dynamics in Diffuse Gliomas during Gliomagenesis and Its Impact on Patient Survival"

_cancers, 2022, doi:10.3390/cancers15010145_

Round 1

Reviewer 1 Report

Intertumoral heterogeneity in GBM is a significant problem for developing effective therapy against this deadly disease, and we have limited characterization of heterogeneity in clinical samples. Authors in this manuscript set to characterize the intertumoral heterogeneity in low-grade GBM by using Flow, FISH, and PCR. The strength of the manuscript is the characterization of the clinical samples. However, one of the significant limitations of this manuscript is that the authors use a single marker to define a specific population within the tumor, as we know that these GBM cells are highly plastic and convert between different cell fates. Authors should consider the following to improve the manuscript.

1.     Please provide a table describing all the patient characteristics.

2.     One of the significant limitations of this manuscript is that the authors use a single marker to define a specific population within the tumor. As we know that these GBM cells are highly plastic and convert between different cell fates. Authors should discuss this limitation in their introduction as well as their discussion.

3.     mRNA expression doesn’t correlate with the protein expression in any given cells. The author should try to complement some of their FISH data for major markers with IHC data.

4.     The conclusion in Fig 4 is inappropriate as it is conducted in cell culture hypoxia conditions. What is the evidence that this will happen within the tumor microenvironment Authors should validate this in clinical samples with IHC with hypoxia markers such as HIF staining and combining with FISH. 

Author Response

First of all, on behalf of the entire team of authors and on my own, I would like to sincerely thank reviewer 1 for the work done. We carefully studied all the comments addressed to the article and tried to take them into account as fully as possible for the revision of our paper. We believe that the comments of the reviewer 1 allowed us to significantly improve our article and make it more understandable and interesting for the widest possible range. The work of reviewer 1 has made an valuable contribution to the formation of a new version of the article. 
Below we provide detailed answers and explanations to all comments and remarks of the reviewer.    

1.    “Please provide a table describing all the patient characteristics”
We strongly agree with the reviewer that the table with epidemiological data of patients improves the perception of the material, in connection with which we have added this table in the appropriate place in the Materials and Methods section on page 3.
2.    “One of the significant limitations of this manuscript is that the authors use a single marker to define a specific population within the tumor. As we know that these GBM cells are highly plastic and convert between different cell fates. Authors should discuss this limitation in their introduction as well as their discussion”
We completely agree with the reviewer regarding the limited value of markers of tumor cell populations. We have clarified this in the Introduction on page 3, line 90, and in the Discussion on page 22, line 1575.
3.    “mRNA expression doesn’t correlate with the protein expression in any given cells. The author should try to complement some of their FISH data for major markers with IHC data”
The reviewer comment is absolutely correct, in connection with which we supplemented our work with an immunofluorescence study for proteome profiling. In the text of the article, we added the method on page 6 in line 691, we also added the corresponding data in simple summary, abstract, results, we also added the corresponding pictures – 1e and 1f, 3g and 3h, 5e, 6e, 8e.
4.    “The conclusion in Fig 4 is inappropriate as it is conducted in cell culture hypoxia conditions. What is the evidence that this will happen within the tumor microenvironment Authors should validate this in clinical samples with IHC with hypoxia markers such as HIF staining and combining with FISH”
We fully agree with the reviewer comment and conducted an additional study of the tumor tissue of patients with increased production of HIF-1α in tissue, comparing the cell-population composition of these tumors with samples without increased production of the hypoxic marker. Appropriate additions have been made to the Materials and Methods section, to the Results section on page 13 and page 19, and figures 4e, 8f has been added.

Reviewer 2 Report

In the current manuscript, entitled” Cell populations dynamics in diffuse gliomas during gliomogenesis and its impact on patient survival”, Nikitin et al., showed that a comparative dynamic analysis of the different cell populations content in diffuse gliomas of different molecular profiles and grades. They found that mesenchymal cell populations appearance arising from oligodendrocyte-progenitor-like cells in the IDH-mutant astrocytomas and oligodendrogliomas. Further, they revealed proneuronal glioma stem cells are the most likely cells source of mesenchymal glioma stem cells in GBM. Finally, they showed mesenchymal-like cells and mesenchymal glioma stem cells have significant effect on relapse free and overall survival.

In general, this article is targeted a very interesting question that intratumoral heterogeneity in diffuse gliomas. The experiments are well designed, and the results are clearly displayed. Most conclusions are supported by their data. I have two main questions before it is published.

1 Please indicate the color code for each gene in the figures with mrFISH. Just showing in method is difficult for readers to follow.

2 For “Cell populations effect on patient prognosis” section, the authors need to point out what is the effect, increased or decreased RFS and OS. The authors need to show the data or reference in this section. They only showed some statistics, but I cannot find where they are coming from.

Author Response

First of all, on behalf of the entire team of authors and on my own, I would like to sincerely thank reviewer 2 for the work done. We carefully studied all the comments addressed to the article and tried to take them into account as fully as possible for the revision of our paper. We believe that the comments of the reviewer 2 allowed us to significantly improve our article and make it more understandable and interesting for the widest possible range. The work of reviewer 2 has made an valuable contribution to the formation of a new version of the article. 
Below we provide detailed answers and explanations to all comments and remarks of the reviewer.    

1.    “Please indicate the color code for each gene in the figures with mrFISH. Just showing in method is difficult for readers to follow”
We do agree that color markers should also be deciphered under the drawings to improve the readability of the figures, and therefore we have added this information in each figure.
2.    “For “Cell populations effect on patient prognosis” section, the authors need to point out what is the effect, increased or decreased RFS and OS. The authors need to show the data or reference in this section. They only showed some statistics, but I cannot find where they are coming from”
This comment is absolutely correct, we agree that data on the effect of cell populations on survival are not presented in sufficient volume. In this regard, we have added a more detailed description in the corresponding section of the Results on page 21, and also added Figure 9 with Kaplan-Meier curves. 
